# Inpatient Weight Restoration Treatment Is Associated with Decrease in Post-Meal Anxiety

**DOI:** 10.3390/jpm11111079

**Published:** 2021-10-24

**Authors:** Magnus Sjögren, Ismail Kizilkaya, Rene Klinkby Støving

**Affiliations:** 1Psychiatric Center Ballerup, 2750 Ballerup, Denmark; ismail.kizilkaya.01@regionh.dk; 2Institute for Clinical Science, Umeå University, 90185 Umeå, Sweden; 3Center for Eating Disorders, Odense University Hospital, Mental Health Services in the Region of Southern Denmark, 5000 Odense, Denmark; rene.k.stoving@gmail.com; 4Research Unit for Medical Endocrinology, Department of Clinical Research, University of Southern Denmark, 5000 Odense, Denmark

**Keywords:** anorexia nervosa, weight restoration treatment, anxiety, meal-intake, post-meal

## Abstract

Objective: Anorexia nervosa (AN) is characterized by weight loss, distorted body image with fear of becoming fat and associated with anxiety, especially in relation to food intake. Anxiety in relation to meals and weight restoration remains a major challenge in the treatment of AN. We examined the effects of inpatient weight restoration treatment on levels of post-meal anxiety using visual analogue scale (VAS) ratings in patients with AN. Materials: Thirty-two patients with AN, all part of the PROspective Longitudinal all-comer inclusion study on Eating Disorders (PROLED) were followed over eight weeks with baseline psychometric measures and weekly VAS anxiety self-scoring. Methods: Apart from the weekly body mass index (BMI) and VAS, patients were characterized at baseline using the Eating Disorder Examination Questionnaire (EDE-Q), Eating Disorder Inventory (EDI), Symptom Check List 92 (SCL-92), Major Depression Inventory (MDI), and Autism Quotient (AQ). Results: The results showed a significant time effect, Wilks Lambda = 0.523, F = 3.12, *p* < 0.05 (power of 0.862), indicating a reduction in VAS scores of anxiety from baseline to week 8. There was no effect of baseline medication or scores of MDI on the results. BMI increased from a mean of 15.16 (week 1) to 17.35 (week 8). In comparison, patients dropping out after only three weeks (*n* = 31) also had a trend toward a reduction in VAS anxiety (ns). Conclusions: Inpatient weight restoration treatment is associated with a decrease in post-meal anxiety in AN, an effect that occurs early and becomes clinically significant in patients who stay in treatment.

## 1. Introduction

Anorexia nervosa (AN) is a multifactorial disease [1] characterized by severe weight loss and distorted body image with fear of becoming overweight [1]. Weight loss in AN is often achieved and maintained primarily by reduced food intake, excessive exercise or purging [2], with severe medical complications as a long-term consequence [3]. In addition, mortality rate in AN is the highest compared to any psychiatric disorder with an almost six-fold increased mortality compared to the normal population [4,5]. 

Evidence-based psychological treatment in adult AN have been graded as weak [1] including enhanced cognitive behavioral therapy, focal psychodynamic psychotherapy, Maudsley model of AN treatment for adults, and specialist supportive clinical management, with none of these having a clear superiority over the other [1]. Frequently, weight restoration therapy is given in the acute stage, having in the short-term been shown to improve prognosis [6].

Several findings link anxiety to AN (e.g., some studies have suggested that anxiety symptoms precede the onset of the eating disorder [7,8]) and that AN and anxiety disorders share a common genetic transmission [9]. Furthermore, individuals with childhood anxiety disorders (e.g., overanxious disorder) and AN are more likely to develop additional anxiety disorders [10], and individuals who had recovered from AN for at least 12 months and who had never before met the criteria for an anxiety disorder, still reported higher levels of anxiety [7]. 

Anxiety in AN may also be secondary to malnutrition [11]. In a food exposure experiment, BMI was negatively correlated with anxiety before meal intake (i.e., the lower an individual’s BMI, the more anxiety-provoking individuals with AN find the thought of and exposure to food to be) [12]. The anxiety may additionally remain even after weight restoration, as shown in a study where pre-meal anxiety in weight-restored patients with AN was significantly higher compared to healthy controls independent of meal type digested [13]. Furthermore, meal related anxiety has been found to be directly correlated to outcome [13,14], and even proposed as a specific target for treatment to improve outcome [13]. Thus, efforts to reduce meal related anxiety in a ward setting, remains a critical prerequisite for being able to achieve better treatment results with fewer relapses.

As the evidence for an effect of weight increase on anxiety is still sparce, and meal-related anxiety is an important prognostic factor [13,14], and since increasing meal-related anxiety may directly counteract treatment efforts, and the influence of several factors such as medication and comorbid depression has not been studied, we aimed to examine levels of post-meal anxiety in patients with AN undergoing weight restoration as inpatients. Based upon clinical observations of inpatients with AN, we expected that anxiety would remain stable or even increase as the patients gained weight, hypothetically because weight gain would trigger the psychopathology of fear of weight gain. 

## 2. Materials and Methods

### 2.1. Participants

Participants included 30 females and two males with ages between 18 and 52 years who all met the ICD-10 criteria for AN at the time of hospital admission [15]. Participants were all enrolled in the PROspective Longitudinal all-comer inclusion study on Eating Disorders (PROLED) study, which commenced in 2016, and is a clinical, longitudinal, 10-year annual follow-up study of Eating Disorders (ED) at the Psychiatric Center Ballerup (PCB). PROLED is approved by the local ethics board (id: H-15012537; addendum 77106) and the data processing board, and the following specifications apply for enrolment after a signed written informed consent:-adult individuals (age 18–65 years);-admitted to the ED unit in Psychiatric Center Ballerup, Denmark; and-a diagnosis of an ED.

Subjects undergoing forced care were excluded from the study. The enrolment rate was 96% in 2016, 74% in 2017, 62% in 2018, and 68% in 2019 at the inpatient unit where this study was conducted. The PCB unit is the only ED unit for treatment of ED in the capitol region of Denmark, having a catchment area of two million inhabitants and an expected prevalence of AN of 2000 individuals.

Patients had undergone medical and psychiatric examination prior to hospitalization, and psychiatric comorbidities and medication were derived from medical records. Relevant psychotropic medication of the participants can be seen in Table 1.

### 2.2. Weight Restoration Treatment

All patients underwent a weight restoration program as inpatients. Meals were provided five times per day during monitoring by trained nurses to ensure proper renourishment. A dietician held individual weekly meetings with each patient to ensure a meal plan that would enable an approximate 1 kg (kg) weight increase per week up to an ideal body weight (IBW) of BMI 20 for women and BMI 21 for men. Weight gain was supported by restrictions in physical activity, monitored meals and post-meal rest. Weekly measures of weight were conducted. All patients had undergone medical and psychiatric examinations and any medical complications were addressed as they were identified. During these eight weeks, there was no formal psychotherapy provided, although individual meetings with psychologists and nurses for supportive reasons were offered to the patients. All patients received vitamins, however, no patient in this study underwent enteral feeding during this current course of treatment. In addition, trained physiotherapists offered a body relaxation program to all patients as part of the clinical inpatient program. An average stay for the ED unit was 10 weeks independent of reason for discharge. 

### 2.3. Clinical and Psychometric Measures

Initial assessments included a complete diagnostic work-up done via a comprehensive diagnostic interview by a psychologist, and medical and psychiatric examinations carried out by either a specialist psychiatrist, or a General Practitioner with special training in EDs. Assessment was accomplished using the Eating Disorder Examination (diagnostic questions; EDE [16] and routine clinical and laboratory assessments to ensure high quality diagnosing of eating disorders (ED) and comorbid disorders. All primary and comorbid diagnoses were validated by a second, independent physician using the ICD-10 checklist [17].

All participants were weighed once a week to calculate BMI (kg/m^2^). The PROLED study includes validated questionnaires that assess the general and specific aspects of ED as well as other psychiatric comorbidities or disorders such as anxiety, depression, and autism. For this study, the Eating Disorder Examination-Questionnaire (EDE-Q) [18,19,20] the Eating Disorder Inventory (EDI) [21,22], the Hopkins Symptom Checklist (SCL-92) [23], the Major Depression Inventory (MDI) [24], and the Autism Quotient (AQ) [25] were used and completed by each participant to characterize the enrolled participants at baseline, both in the psychopathology of their ED and the degree of psychiatric comorbidity. Only validated Danish versions of the instruments were used. 

The EDE-Q [19] is a self-report questionnaire that was developed from the investigator-based interview instrument, the Eating Disorder Examination (EDE). The EDE-Q is designed to measure the broad range of the specific psychopathology of eating disorders by measuring the present state of the eating behavior and attitudes of subjects over the previous 28 days. Eating behavior and attitudes of subjects are assessed using a 7-point scale (from 0 to 6). Points are measured in terms of the number of days the subjects experience a certain eating disorder behavior, where high EDE-Q scores indicate high levels of ED pathology. In the Danish version of the EDE-Q, subjects are asked to answer a total of 28 items. Four sub-scales may be derived from its ratings: dietary restraint, eating concern, weight concern, and shape concern. A global scale and subsequent global score can be calculated as well as the average of the four sub-scale scores.

The original EDI [22] is a self-report questionnaire that consists of 64 items for the assessment of psychological and behavioral traits common for eating disorders. Since its development, revisions have been made to the EDI, and in this study, the third and latest revision was used. The EDI-3 [21] consists of 91 questions and each item scores 0–4 points. For all participants, a total sum score was calculated based on each item score. The EDI-3 assesses the psychological traits and symptoms relevant to the development and maintenance of eating disorders. For each item, the participants’ possible answers were “always”, “mostly”, “often”, “sometimes”, “rarely”, or “never”. 

In this study, the validated Danish version of the 92-item of the Hopkins Symptom Checklist [23] was answered by each participant. Each item on the checklist was ranked in five levels from “totally agree” to “do not agree at all”. The SCL-92 can be used to assess different psychiatric symptoms such as depression, anxiety, interpersonal sensitivity, somatization, obsession/compulsion, paranoid ideation, psychoticism, and more. The anxiety and depression subscales of the SCL-92 were used to assess baseline levels of depression and anxiety. 

The MDI [24] is a unidimensional self-rating instrument that covers the ICD-10 symptoms of depression for the past 14 days. Each item on the MDI gives a score between 0 and 5 on a Likert scale and the scores are summed up with a theoretical score range between 0 to 50, where the higher the total score, the more severe the depression. 

The AQ [25] is made up of 10 questions assessing five different areas: social skill, attention switching, attention to detail, communication, and imagination. Participants are supposed to answer each of the 50 questions with “definitely agree”, “slightly agree”, “slightly disagree”, or “definitely disagree”, according to the degree they believe they exhibit the behavior. A combined score is then calculated. 

### 2.4. Measurements of Anxiety

During the eight weeks of the study, while spending time in the relaxing environment, participants were asked once a week post-meal on every Thursday at 1 pm local time to rate their level of anxiety. Ratings were measured using a visual analogue scale (VAS) with a scoring range from 1–10, with 1 representing no anxiety and a score of 10 representing the highest imaginable anxiety experienced.

### 2.5. Statistical Analysis

Sample size was calculated assuming a baseline mean of 5 in VAS score, an endpoint VAS mean of 6, and a SD of 1, setting alpha to 0.5 and a power of 80%. The UBC (University of British Colombia, Vancouver, Canada) statistical online module was used (https://www.stat.ubc.ca/~rollin/stats/ssize/n2.html (accessed on 5 April 2021)), which provided a power of 16 individuals at baseline and endpoint. 

All analyses were conducted using the Statistical Package for Social Sciences (SPSS) Version 27 for Macintosh computer. The distribution of the included data was assessed using the Kolmogorov–Smirnoff test and any outliers and/or missing data were excluded by case. VAS anxiety scores were normally distributed, and thereby parametric statistics were applied.

Clinical characteristics were described as the means and standard deviation for age, duration, and BMI. Gender distribution was described in numbers and percentage. Correlations, using Pearson correlation, between VAS scores and potential co-variates such as age and duration of disease were calculated at all time-points, and if correlated, were included in the ANOVA (see below).

A one-way repeated measures analysis of variance (ANOVA) was conducted to evaluate the null hypothesis that there is no change, or an increase, in the participants’ anxiety scores post-meal intake from week 1 to week 8 of the inpatient weight restoration treatment. Results were presented as a time effect using Wilks Lambda, F-statistics, and power.

## 3. Results

### 3.1. Participants’ Baseline Clinical Characteristics and ED Psychopathology

The descriptive data in the sample are summarized in Table 2. The 32 patients diagnosed with typical or atypical AN had a mean age of 25.63 years (SD = 8.67), and a mean baseline BMI of 15.16 kg/m^2^ (SD = 2.07). Thirty of the 32 patients were women (94%), and the mean duration of illness across the entire sample was 8.00 years (SD = 9.2). Ten patients received anti-depressive medication, eight received antipsychotic medication, and five received anxiolytic medication during the time of the study (see Table 1). During the eight weeks, mean BMI increased from 15.16 (SD 2.7) to 17.35 (SD 1.79). 

Table 2 also shows the mean scores of participants for the EDE-Q, EDI, and SCL-92 scores of anxiety and depression, and the scores of MDI and AQ for characterization of the sample. Overall, patients may be described as moderate-severe to severely affected by their AN.

### 3.2. Anxiety Outcomes 

Neither age nor duration of disease were correlated with VAS-scores of anxiety at any timepoint during the 8 weeks, and were therefore not included in the repeated measures ANOVA. 

The results of the repeated measures ANOVA indicated a significant time effect, Wilks Lambda = 0.523, F = 3.12, *p* < 0.05 with an observed power of 0.862. There was no effect of baseline medication (e.g., antidepressants, antipsychotics, or anxiolytics) usage, nor any influence of the baseline scores of depression, as measured by MDI, on the results of the repeated measures ANOVA. 

Follow-up comparisons indicated that the following pairwise comparisons were significant; between week 8 and 1 (*p* < 0.05) and week 7 and 1 (*p* < 0.05) (Figure 1).

A significant reduction in mean anxiety scores was found when week 8 and week 7 were compared to week 1. There was no significant reduction in mean anxiety scores when any other week were compared to week 1. Figure 1 shows a significant reduction in post-meal anxiety scores of participants from week 1 to week 8, when an ANOVA was conducted. Baseline medication with anti-depressive medication, antipsychotics, or anxiolytics and baseline depression scores measured with MDI had no effect on the results.

To investigate the effect of gender, we re-ran the ANOVA with only women included, which yielded the same results (Wilks Lambda 0.52; F = 3.1, *p* = 0.02; observed power = 0.85). Since only two men were included, no similar statistical calculation was carried out for men only.

To enable a comparison with patients who dropped-out early from the VAS assessments, either due to being moved to other care (e.g., daycare or ambulant care: original sample included the 32 patients completing eight weeks; *n* = 61) or were discharged from the department for other reasons, there was a non-significant reduction already after three weeks in VAS-anxiety (Figure 2), and in parallel, a weight increase in the whole group (Figure 3). In addition, comparing levels of post-meal anxiety at three weeks between early drop-outs and those who stayed in treatment for eight weeks revealed no difference between the groups (F = 0.006, *p* = 0.94; Figure 2). 

## 4. Discussion

In this study on patients with moderately severe to severe AN, being under inpatient weight restoration treatment, it was found that levels of post-meal anxiety decreased early on and became clinically significant after 7–8 weeks. The effect applied both to those who dropped out early from treatment for various reasons and those who remained in treatment for eight weeks. This refuted our clinical presumption of an increase in anxiety during weight gain; however, adheres to what has been described by others [12]. It also underscores the relation between being extremely underweight and anxiety. In addition, there was no influence of baseline medication or baseline scores of depression on the improvements in anxiety, which stresses that the reduction in anxiety scores is independent of these factors. This finding provides support to inpatient weight restoration in reducing symptoms associated with AN.

The results of this study also stresses the relevance of measuring meal related anxiety. Previous studies have found meal related anxiety to be related to outcome, both in underweight patients with AN [14] and in weight restored AN [13], and at least one study found that therapeutic interventions in relation to meal intake may influence meal-related anxiety [26]. The results of this study add to the number of studies including meal related anxiety as an outcome for intervention trials.

Patient were hospitalized due to moderate to severe AN, which was reflected in the high scores on all four EDE-Q subscales and total EDI score [20]. Current literature suggests that a significant portion of the individuals with AN suffer from additional psychiatric disorders or comorbidities with depression and anxiety being the most prevalent of all [27,28,29,30]. Indeed, the baseline scores of depression and anxiety with SCL-92 and MDI indicated moderate to severe depression, whereas the low scores in AQ ruled out that autism spectrum disorders would have impacted the results.

The current study must be seen in light of the following limitations. First, only hospitalized patients diagnosed with AN were enrolled in this study. Hospitalization of patients with AN usually involves severe eating disorder psychopathology and comorbidities. Therefore, this study cannot inform on how anxiety levels change in non-hospitalized patients with AN undergoing weight restoration. In addition, the results apply to patients who remain in treatment, and little can be inferred to patients with mild to moderate stages of AN. Furthermore, although the study was sufficiently powered to detect an effect in VAS scores of anxiety over eight weeks, including a larger number of patients with AN may have identified an earlier effect as well as the effect of influencing factors such as medication. Only 32 out of 61 admitted patients were eligible for eight-week assessment. Pre-term discharge and drop out are a very well-known problem in treating AN, where the subtype binge and/or purging is of importance [31]. Another limitation is that all individuals included were from the same geographical region and it may be that there are differences relating, for example, to ethnicity where differences in body image disturbance may influence the results [32,33]. Moreover, there were only two males included in the current study, and they remained throughout the eight weeks. However, any effects of gender in this study, since there were only two males included, must be interpreted with caution.

Anxiety was assessed by a simple and manageable VAS, which has been validated and proven to be a reliably and sensitive method in several different patient populations (e.g., in pre-anesthetic situations and at the dentist) [34,35]. However, anxiety remains a multifaceted complex emotion, and in AN, it can probably be nuanced by future studies with multidimensional measurement methods.

In the present study, we related alterations in post meal anxiety to the parallel increase in BMI, because weight phobia and refusal of treatment is an essential challenge in all treatment of AN. However, obviously, there are other factors which influence anxiety during inpatient treatment. For example, during the first few weeks, the environmental therapy could gradually make the patients more comfortable during hospitalization, regardless of the changes in BMI.

## 5. Conclusions

The current study found that patients with AN showed an improvement in anxiety during inpatient weight restoration treatment, with no influence from baseline medication or depression. This gives support to inpatient weight restoration treatment to treat symptoms associated with AN in individuals that are moderately-severe to severely affected by the disorder.

## Figures and Tables

**Figure 1 jpm-11-01079-f001:**
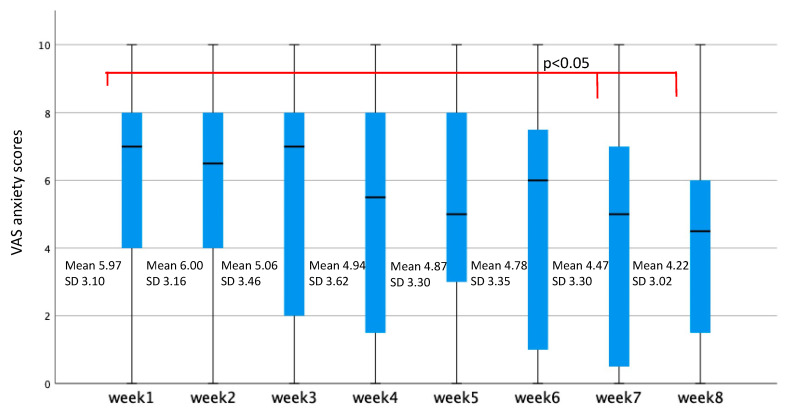
Box and whisker plots indicating median, 95% CI and extremen values of VAS anxiety scores over eight weeks in anorexia nervosa. Corresponding means and SD are inserted for comparison.

**Figure 2 jpm-11-01079-f002:**
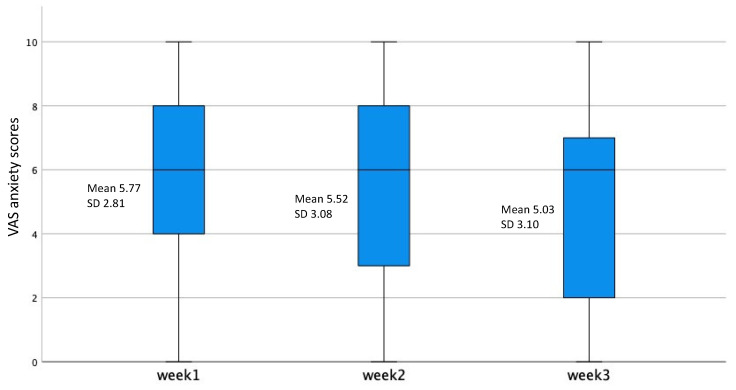
Box and whisker plots indicating median, 95% CI and extreme values of VAS anxiety scores over three weeks for individuals with anorexia nervosa dropping out after three weeks. Corresponding means and SD are inserted for comparison.

**Figure 3 jpm-11-01079-f003:**
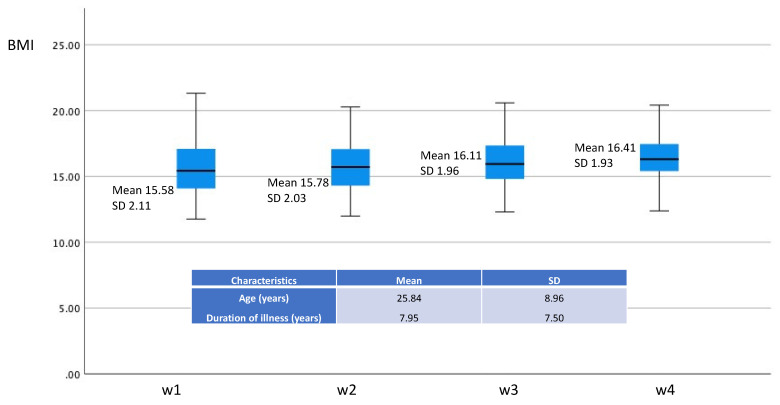
Box and whisker plots indicating median, 95% CI and extreme values of BMI in the drop-out group during four weeks of weight restoration therapy. Corresponding means and SD are inserted for comparison.

**Table 1 jpm-11-01079-t001:** The number (*n*) of patients (total *n* = 32) receiving psychotropic medication during the study. ATC: Anatomical Therapeutic Chemical Classification System.

Category	Frequency (*n*)	%
**Antidepressant**	10	31.3
ATC N06AB		
**Antipsychotic**	8	25
N05AX, N05AF		
**Anxiolytic**	5	15.6
ATC N05BA N06AB		

**Table 2 jpm-11-01079-t002:** Participants’ (total *n* = 32, female = 30 and male = 2) baseline clinical characteristics and BMI.

Characteristics	Mean	SD
Age (years)	25.63	8.67
Duration of illness (years)	8.00	9.2
BMI week 1	15.16	2.07
BMI week 2	15.40	2.06
BMI week 3	15.80	1.95
BMI week 4	16.20	1.86
BMI week 5	16.45	1.88
BMI week 6	16.84	1.82
BMI week 7	17.08	1.82
BMI week 8	17.35	1.79
EDEQ-Restraint	3.49	1.76
EDEQ-Eating concern	3.37	1.58
EDEQ-Shape concern	4.65	1.50
EDEQ-Weight concern	4.29	1.69
EDEQ-Global score	3.95	1.48
EDI	170.28	67.23
SCL-A_14_	2.00	0.92
SCL-D_16_	2.26	0.96
MDI	33.03	9.61
AQ	19.98	8.89

## Data Availability

The datasets used in this study are available from the corresponding author.

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
