# Peer review of "Inpatient Weight Restoration Treatment Is Associated with Decrease in Post-Meal Anxiety"

_jpm, 2021, doi:10.3390/jpm11111079_

Round 1
Reviewer 1 Report
This article by Sjogren et al describes a cohort of individuals undergoing weight restoration treatment. The manuscript is well written and does a good job summarizing the importance of the study. The results are however a bit shallow, needing to show better the data to allow readers to better understand the data. The results themselves are shorter than any other section, which to me is normally a red flag. The authors should expand on their data insights of the cohort.
Make sure all abbreviations are defined at first use.
Lines 212-214 need to be removed or the data available somewhere. In the age of open data, these statements are no longer acceptable.
Tables 3-5 would be better shown as a box and whisker plot. Table does not allow one to understand outliers and the data structure. Otherwise all data needs to be available in a supplemental file.
The discussion needs to include limitation on cultural aspects of eating. It is fair to assume these patients have shared cultural upbringing in that enrollment was from a single site. Thus, how do the authors think this would apply to broader ethnicities and cultures?
Author Response
We thank this peer reviewer for valuable feedback and suggestions for improvements. We have updated the manuscript accordingly. Below are a list of our responses to the comments. Peer reviewer 1 This article by Sjogren et al describes a cohort of individuals undergoing weight restoration treatment. The manuscript is well written and does a good job summarizing the importance of the study. The results are however a bit shallow, needing to show better the data to allow readers to better understand the data. The results themselves are shorter than any other section, which to me is normally a red flag. The authors should expand on their data insights of the cohort.- we have added data on gender differences in anxiety over 8 weeks
- we have added data on the impact of age on the results
- we have box and whisker plots as requested below.
- we would humbly like to emphasize that also all tables and box-whisker plots belong to the results section. In the tables, a lot of data are described.
- we have reviewed and updated on the abbreviations throughout the manuscript. we have also added a list of abbreviations in the end of the manuscript.
- we agree and have added results from the analysis.
- we have added box and whisker plots to display VAS anxiety scores and confidence intervals.
- we agree and have added a section on how cultural differences may influence the results. We have stressed that the psychopathology of AN likely is the same while that different ethnic groups, for example asian individuals with AN, may differ from this caucasian sample of individuals with AN, in anxiety scores over weight restoration treatment.
Reviewer 2 Report
The study by Sjögrenet al is interesting. My concerns are about samples stratification. The authors groups men with female and quite different in age.
Would the reysults show the same significance when the patients are grouped differently.?
Author Response
We thank this peer reviewer for valuable feedback and suggestions for improvements. We have updated the manuscript accordingly. Below are a list of our responses to the comments.
Peer reviewer 2 The study by Sjögren et al is interesting. My concerns are about samples stratification. The authors groups men with female and quite different in age.- we have adjusted for age in the repeated measures ANOVA analysis and the results are the same. This has been included in the results section.
- we have also compared men and women and the results are the same (see added information in the results section)
- The results did not change after adjusting for age, and when comparing women to me.
Round 2
Reviewer 2 Report
I could not understand reviewers reply and the way they calculate the data. What is the number of male patients? Compared to what?
Author Response
Dear Peer reviewer,
we thank this reviewer for valuable comments and agree that the description of the statistics was unclear.
1) The number of male patients is 2. In addition, there were 30 females included. This is described on line 70, under material and methods
2) we agree that a description of the calculation was unclear and this has now been added in the Statistical and the results sections respectively, which no reads:
Statistics:
"Correlations, using Pearson correlation, between VAS scores and potential co-variates such as age and duration of disease were calculated at all time-points, and if correlated, included in the ANOVA (see below)."Results:
"To investigate the effect of gender, we re-run the ANOVA with only women included which yielded the same results (Wilks Lambda 0.52; F=3.1, p=0.02; observed power =0.85). Since only two men were included, no similar statistical calculation was made for men only."
Consequently, these two sentences were excluded since they were inadequate:
Comparing men and women, there were no effect of time on VAS scores on anxiety over 8 weeks, Wilks Lambda = 0.770, F=1.02, p=0.442 with an observed power of 0.344 meaning that the reduction in anxiety was following the same pattern in men and women.
There was no effect of age on the results on the repeated measures ANOVA (F=0.687, p=0.752).